# Fractalkine/CX3CL1 in Neoplastic Processes

**DOI:** 10.3390/ijms21103723

**Published:** 2020-05-25

**Authors:** Jan Korbecki, Donata Simińska, Klaudyna Kojder, Szymon Grochans, Izabela Gutowska, Dariusz Chlubek, Irena Baranowska-Bosiacka

**Affiliations:** 1Department of Biochemistry and Medical Chemistry, Pomeranian Medical University in Szczecin, Powstańców Wielkopolskich 72 Av., 70-111 Szczecin, Poland; jan.korbecki@onet.eu (J.K.); d.siminska391@gmail.com (D.S.); szymongrochans@gmail.com (S.G.); dchlubek@pum.edu.pl (D.C.); 2Department of Anaesthesiology and Intensive Care, Pomeranian Medical University in Szczecin, Unii Lubelskiej 1 St., 71-281 Szczecin, Poland; klaudynakojder@gmail.com; 3Department of Medical Chemistry, Pomeranian Medical University in Szczecin, Powstańców Wlkp. 72 Av., 70-111 Szczecin, Poland; izagut@poczta.onet.pl

**Keywords:** CX3CL1, fractalkine, CX3CR1, cancer, chemokine, metastasis

## Abstract

Fractalkine/CX3C chemokine ligand 1 (CX3CL1) is a chemokine involved in the anticancer function of lymphocytes—mainly NK cells, T cells and dendritic cells. Its increased levels in tumors improve the prognosis for cancer patients, although it is also associated with a poorer prognosis in some types of cancers, such as pancreatic ductal adenocarcinoma. This work focuses on the ‘hallmarks of cancer’ involving CX3CL1 and its receptor CX3CR1. First, we describe signal transduction from CX3CR1 and the role of epidermal growth factor receptor (EGFR) in this process. Next, we present the role of CX3CL1 in the context of cancer, with the focus on angiogenesis, apoptosis resistance and migration and invasion of cancer cells. In particular, we discuss perineural invasion, spinal metastasis and bone metastasis of cancers such as breast cancer, pancreatic cancer and prostate cancer. We extensively discuss the importance of CX3CL1 in the interaction with different cells in the tumor niche: tumor-associated macrophages (TAM), myeloid-derived suppressor cells (MDSC) and microglia. We present the role of CX3CL1 in the development of active human cytomegalovirus (HCMV) infection in glioblastoma multiforme (GBM) brain tumors. Finally, we discuss the possible use of CX3CL1 in immunotherapy.

## 1. Introduction

It is estimated that 18.1 million people worldwide suffer from cancer each year, with 9.6 million of these ending in death [1]. In most developed countries, as cancer constitutes the most common cause of death in people under 70, neoplastic processes are being intensively studied in order to develop better and more effective therapeutic approaches. Twenty years ago, research focused mainly on changes occurring within neoplastic cells [2], especially mutations and intracellular signaling in an isolated neoplastic cell. This approach has changed along with advances in research methods, showing numerous links between cancer cells and tumor niche cells (cancer-associated fibroblasts (CAF), tumor-associated macrophages (TAM), myeloid-derived suppressor cells (MDSC), microglia, neutrophils, regulatory T cells (T_reg_) and many others) [3,4]. These relationships are significant in the development of cancer and depend on the recruitment of cells to a cancer niche and intercellular signaling between these cells, both of which involve chemokines [5].

To date, more than 50 chemokines have been described and subsequently divided into four families, depending on the occurrence of a specific motif in the amino acid sequence at the N-terminal end. One such chemokine that plays an important role in the cancer process is CX3C chemokine ligand 1 (CX3CL1) along with its receptor CX3C chemokine receptor 1 (CX3CR1). This chemokine stimulates cancer cell proliferation [6,7,8] and participates in angiogenesis [9,10,11], apoptosis resistance [12] and cancer cell migration [13,14,15,16]. CX3CL1 also participates in the recruitment of cells to a cancer niche [17,18]. The most characteristic feature of this chemokine is its participation in organ-specific metastasis of several cancers according to ‘the seed and soil’ hypothesis [15,19,20,21].

Since the discovery of CX3CL1 more than 20 years ago [22], a number of studies on its role in cancer processes have been undertaken. However, the fragmentary nature of those reports has resulted in controversies regarding the properties and activity of this chemokine. Therefore, this paper attempts to present a comprehensive review of research on its pro- and anticancer functions.

## 2. CX3CL1 Protein

CX3CL1 is a chemokine with a cysteine signature motif -Cys-X-X-X-Cys- at the N-terminal end and is the only known representative of the δ-chemokine family. It was first discovered in 1997 [22] and named neurotactin [23] and fractalkine [24], with the latter name currently in use. CX3CL1 shows high expression in the brain, where it is synthesized by neurons and has neuroprotective functions [25,26,27,28]. High expression of CX3CL1 is also found in the kidneys, lungs [29], bones [14] and blood vessels [18,30,31,32,33].

There are three main CX3CL1 transcripts (1.8 kb, 4 kb and 7 kb long), which are likely created by alternative splicing [29]. CX3CL1 is synthesized as a 50–75 kDa precursor protein [34]—as a 373-amino acid polypeptide in humans [22] and a 395-amino acid polypeptide in mice [29]. The synthesized polypeptide is glycosylated, after which CX3CL1 is built into the cell membrane and occurs on cells as 100 kDa type I transmembrane glycoprotein [23,34]. The membrane-attached form of CX3CL1 (mCX3CL1) has a short intracellular and transmembrane domain. The most significant part of mCX3CL1 is the chemokine domain, which connects to the transmembrane domain via a mucin-like stalk [22], which does not affect the properties of mCX3CL1 but only facilitates presentation of the chemokine domain to other cells [35]. In this form, CX3CL1 acts as an adhesion protein for cells with CX3CL1 receptor expression [36], such as NK cells, CD3^+^ T cells, monocytes, dendritic cells and granulocytes [24,26].

CX3CR1 expression is elevated in CD4^+^ and CD8^+^ T cells in exposure to interleukin 2 (IL-2). However, the expression of CX3CL1 in blood vessels is elevated by proinflammatory cytokines interferon-γ (IFN-γ), tumor necrosis factor α (TNF-α) and interleukin 1β (IL-1β) [18,30,31,32,33]. Thanks to these properties, CX3CL1 participates in the adhesion and recruitment of immune system cells to areas of a strong immune response [37]. However, this mechanism also occurs in neoplastic processes, more precisely during metastasis. Cancer cells that circulate in the blood can express CX3CR1, which allows them to adhere to mCX3CL1 in blood vessels and, consequently, to migrate to organs distant from the parent tumor [14,19].

mCX3CL1 may be subject to proteolytic cleavage into soluble forms of 85 kDa CX3CL1 (sCX3CL1) [34]. Thus far, four enzymes involved in this process have been identified: cathepsin S [38], matrix metalloproteinase (MMP)-2 [39], a disintegrin and metalloproteinase 10 (ADAM10) [39,40,41] and tumor necrosis factor-α converting enzyme/a disintegrin and metalloproteinase 17 (TACE/ADAM17) [34,39,42,43]. ADAM10 is involved in the constitutive CX3CL1 cleavage [40], while TACE/ADAM17 is involved in CX3CL1 cleavage induced by such factors as phorbol 12-myristate 13-acetate (PMA) [34,42] or proinflammatory cytokines [43]. However, in hCMEC/D3 brain endothelial cells [41] and human astrocytes [44], ADAM10 is induced by proinflammatory cytokines to mediate sCX3CL1 shedding.

## 3. CX3CR1: Signal Transduction

CX3CR1 is the receptor for CX3CL1 [45]. It belongs to seven-transmembrane domain G protein-coupled receptors. Its expression occurs in microglia, where it plays an important role in the interaction of neurons with these cells [25,26,27]. CX3CR1 is also expressed on immune system cells: NK cells, CD3^+^ T cells, monocytes, dendritic cells and granulocytes [24,26]. Expression of mCX3CL1 occurs in vessel walls and is increased by proinflammatory cytokines [18,30,31,32,33]. Thanks to the increased CX3CL1 expression, immune system cells can perform adhesion and transendothelial migration to the sites of strong immune reactions [36].

When mCX3CL1 is bound to CX3CR1, G-protein is not activated. In turn, sCX3CL1 activates its own receptor which in turn activates the G protein coupled with this receptor. The signal transduction is sensitive to pertussis toxin [46], as it is dependent on G_i_. Activating CX3CR1 activates multiple signaling pathways (Figure 1). In particular, it causes Ca^2+^ mobilization [7,25], activation of extracellular signal-regulated kinase (ERK) mitogen-activated protein kinase (MAPK) and phosphatidylinositol-4,5-bisphosphate 3-kinase (PI3K) → Akt/protein kinase B (PKB) pathways [9,46,47], Src → focal adhesion kinase (FAK) pathway [14,16] and janus tyrosine kinase 2 (JAK2) → signal transducer and activator of transcription 3 (STAT3) pathway [48]. The activation of the PI3K→ Akt/PKB pathway can activate nuclear factor κB (NF-κB), and thus, increases apoptosis resistance; additionally, in an autocrine manner, it causes an increase in CX3CL1 expression [12,49]. JAK2 → STAT3 participates in epithelial-to-mesenchymal transition (EMT). On the other hand, activation of ERK MAPK may increase the protein expression of hypoxia-inducible factor-1α (HIF-1α), and thus, supports angiogenesis [10].

It seems that most of the pathways activated by CX3CL1, especially the Src → FAK and PI3K → Akt/PKB pathways, depend on direct activation of epidermal growth factor receptor (EGFR)/ErbB1 and ErbB2 [7,15,50]. The JAK2 → STAT3 pathway also highly likely depends on the activation of EGFR, as similar signal transduction occurs on breast cancer cells [51]. The activation of EGFR/ErbB1 and ErbB2 by CX3CR1 occurs through shedding and releasing of amphiregulin, epiregulin, heparin-binding EGF-like growth factor (HB-EGF) and transforming growth factor α (TGF-α), all of which are EGFR/ErbB1 and ErbB2 activators [7,50,52]. In the signal transmission via CX3CR1, TACE/ADAM17 is responsible for releasing TGF-α [52]. TACE/ADAM17 potentially releases all ligands of the EGFR family [53], and therefore, it may be that only this proteinase is activated by CX3CR1. Thus far, there has been only one report showing the significance of TACE/ADAM17 in the activation of EGFR receptors by CX3CR1 [52].

## 4. The Anticancer Response of the Immune System: The Role of CX3CL1

One of the ‘hallmarks of cancer’ is the genetic instability of a cancer cell [4], which leads to the formation of new antigens triggering an immune response. Antigens allow the elimination of the cancer at an early stage of development, or to more or less effectively fight the developing tumor. An important factor in such a response is CX3CL1, participating in the anticancer response in multiple ways.

The anticancer response is characterized by increased production of IFN-γ, IL-1β and TNF-α – pro-inflammatory cytokines that increase mCX3CL1 expression in vessel walls [31,54] by the activation of NF-κB and specificity protein 1 (Sp1) [30,32]. In addition, TNF-α activates p38 MAPK in blood vessel cells which then activates HuR—a protein increasing the stability of CX3CL1 mRNA and therefore CX3CL1 protein level [33]. Finally, an increase in CX3CL1 expression in a cancer cell is also induced by genetic stress caused by the accumulation of mutations that activate p53 [55].

Due to the fact that mCX3CL1 acts as an adhesion protein for cells with CX3CR1 expression, it causes the retention of immune system cells on vessel walls, close to the site of the inflammatory reaction [56,57,58], enabling trans-endothelial migration of these cells (Figure 2). sCX3CL1 is a chemoattractant for NK cells and dendritic cells due to CX3CR1 expression on these cells [24,26,59]. The expression of this receptor is also seen on CD8^+^ T cells and activated CD4^+^CD45RO^+^ T cells [60]. However, there is no expression of CX3CR1 in eosinophils and neutrophils, and therefore, CX3CL1 does not act directly on these cells [59]. An increase in sCX3CL1 expression in the cancer microenvironment allows the chemotaxis of all the aforementioned cells with CX3CR1 expression towards the cancer niche, where they exert an anticancer effect, with NK cells and CD8^+^ T cells the most significant in the direct anticancer action of CX3CL1 [61,62,63].

In the tumor niche, the CX3CL1-CX3CR1 axis plays an important anticancer role as it allows for the migration of immature dendritic cells to the cancer cell using CX3CL1 expression [64]. This contributes to the adhesion of these two cells with each other and the maturation of dendritic cells. Mature dendritic cells show CX3CL1 expression, and in the tumor niche, this enables the migration of NK cells [65] and T cells [60] to mature dendritic cells. mCX3CL1 expression on mature dendritic cells enables their adhesion to NK cells or T cells along with a stronger subsequent activation of the latter. Additionally, the CX3CL1-CX3CR1 axis allows NK cells to adhere to neoplastic cells, which allows cancer cells to be more effectively killed by NK cells [66,67].

CX3CL1 plays an important role in the anticancer immune response [68]. An increase in CX3CL1 expression in a tumor is correlated with the anti-cancer response of the immune system—the accumulation of anticancer CD4^+^ T cells, CD8^+^ T cells, dendritic cells and NK cells in a tumor [61,64,69,70,71,72,73,74,75,76,77,78]. For this reason, an increase in CX3CL1 expression in the tumor improves the prognosis for patients with breast carcinoma [74], colorectal cancer [71,79], gastric adenocarcinoma [73], glioma [78], lung adenocarcinoma [80] and soft tissue sarcomas [81]. It is even postulated using gene therapy in transferring CX3CL1 to cancer cells [72,82]. Out of two CX3CL1 isoforms, i.e., mCX3CL1 and sCX3CL1, the latter seems to have a strong anticancer effect [83]. mCX3CL1 has shown a weak anticancer effect, e.g., in a study on the metastasis of C26 colon cancer cells to the lung and liver, and a strong anticancer effect in a study performed in a subcutaneous tumor model [83].

CX3CL1-dependent immune processes can be offset by some mechanisms present in the tumor. For example, cancer cells increase the production of transforming growth factor β (TGF-β), especially in advanced stages of cancer [84,85]. This anti-inflammatory cytokine, playing an important role in the development of cancer [86], reduces the expression of CX3CL1 in cancer cells [87] and cancels the action of pro-inflammatory cytokines on CX3CL1 expression in vessel walls [54]. TGF-β has also a direct effect on NK cells by increasing the expression of miR-27a-5p in these cells, which causes a decrease in CX3CR1 expression, and thus, affects CX3CL1-dependent migration of these cells to the tumor niche [88,89]. It should be stressed, however, that the influence of cancer is not local, as the secretion of various factors by a tumor into the blood causes a general dysfunction of the immune system [90,91].

Another process that interferes with the CX3CL1-CX3CR1 axis is hypoxia. In a growing tumor, chronic hypoxia occurs because at some point the growth of the tumor is not matched by the growth of blood vessels feeding the tumor [92]. The tumor also involves another type of hypoxia, known as cycling (intermittent, transient) hypoxia. It consists of fluctuations of oxygen levels, which—depending on the type of cancer—results in alternately occurring hypoxia and reoxygenation from 15 min [93] to several hours [94,95]. This is associated with the abnormalities of blood vessels which do not have a hierarchical structure and show some leakiness due to weak connections between the cells [96,97,98,99]. Chronic hypoxia in a tumor impairs the normal immune response [100,101]; in particular, it abolishes the effect of proinflammatory cytokines on CX3CL1 expression [31], which inhibits the recruitment of anticancer immune system cells into the tumor niche. However, in cycling hypoxia, vascular cells show increased response to proinflammatory cytokines [102].

Another pro-cancer mechanism in a tumor is the reduction in CX3CL1 expression in a cancer cell, in particular via interfering with the functioning of p53, mainly by a mutation of the *TP53* gene [103,104,105]. The *CX3CL1* gene promoter is regulated by p53, and thus, the mutation in the *TP53* gene reduces the expression of this chemokine in the cancer cell [55]. A neoplastic cell also shows an increase in the expression of miR-561-5p, which reduces the expression of CX3CL1 [106]. Another mechanism behind the reduction in the amount of mCX3CL1 on a cancer cell is an increase in the expression of ADAM10 and TACE/ADAM17, enzymes releasing sCX3CL1 [107,108], which results in a reduction in mCX3CL1 on a cancer cell.

CX3CL1 plays an important role in the elimination of neoplastic cells by the immune system. For this reason, many studies show a correlation between increased expression of CX3CL1 and the number of lymphocytes in the tumor and positive patient prognosis [71,73,78,79,80,81,109]. However, during the preparation of this review, we found 10 papers available in PubMed (https://www.ncbi.nlm.nih.gov/pubmed) that concerned correlations between survival rates and CX3CL1 expression level in tumors of various cancers (Table 1). Among them, four studies show that increased expression of this chemokine in the tumor gives a worse prognosis for the patient, especially those with breast cancer [76], grades III-IV brain tumors [110], lung adenocarcinoma among tobacco smokers [111] and pancreatic ductal adenocarcinoma [112]. An increase in sCX3CL1 concentration in the blood was also associated with a worse prognosis in colorectal cancer [113]. All this shows the ambiguous effect of CX3CL1 in cancer. According to The Human Protein Atlas (https://www.proteinatlas.org/) [114], a greater expression of CX3CL1 is associated with a worse prognosis in pancreatic cancer, which is consistent with the aforementioned results of Xu et al. [112], which indicates that CX3CL1 plays an important role in the development of this particular cancer (Table 2).

Another important issue is the effect of CX3CR1 expression in a tumor on the prognosis of cancer patients. An increase in CX3CR1 expression alone may indicate an increase in the number of the cells of the immune system in a tumor and an active anti-tumor response [24,26,59,60,61,62,63]. Another reason for the increase in CX3CR1 expression in a tumor is an increase in the expression of this protein on a tumor cell [19,20,79]. In the absence of CX3CL1, this has a negative effect because if cancer cells get into the bloodstream a site-specific metastasis occurs, as described later in the article. To sum up, the increase in CX3CR1 expression in a tumor may have an either anti-cancer and pro-tumor effect, depending on the cells that express this protein.

Increased expression of CX3CL1 and CX3CR1 in a tumor is both anti-cancer [24,26,59,60,61,62,63] and pro-cancer [6,7,8,9,10,11,12,13,14,15,16,17,18,19,20,21]. Nevertheless, relatively little is known about the effect of the co-expression of these two proteins in a tumor. Increased expression of CX3CL1 causes homing of anti-tumor cells of the immune system that express CX3CR1 [24,26,59,60,61,62,63]. This increases the expression of this receptor protein due to changes in the cellular composition of the tumor. If tumor cells express mCX3CL1 and CX3CR1 simultaneously, these cells will stick together [79]. This significantly impedes the migration of cancer cells. For these reasons, increasing the expression of CX3CL1 and CX3CR1 in the tumor simultaneously improves the prognosis of patients with cancers such as colorectal cancer [79] and hepatocellular carcinoma [109]. Nevertheless, in the tumor, mCX3CL1 may be cleaved by proteinases such as ADAM10 or TACE/ADAM17 [107,108], which releases the chemokine domain and thereby reduces the level of mCX3CL1 on a tumor cell. For this reason, cancer cells no longer adhere to each other via CX3CR1. The receptor itself is activated by sCX3CL1 [46]. This demonstrate the pro-tumor effect of the CX3CL1-CX3CR1 axis, as discussed further in this paper [6,7,8,9,10,11,12,13,14,15,16,17,18,19,20,21]. Probably for this reason, the simultaneous increase in CX3CL1 and CX3CR1 expression would lead to a worse prognosis in patients with pancreatic ductal adenocarcinoma [112].

## 5. Effects on Cancer Cell Proliferation and Apoptosis Resistance

One of the best known ‘hallmarks of cancer’ is the uncontrolled proliferation and apoptosis resistance of cancer cells [2,4]. CX3CL1 stimulates the proliferation of these cells, as has been shown in breast cancer cells [7], gastric cancer cells [115], prostate cancer cells [8] and ovarian carcinomas [6,116]. However, it seems that this effect does not depend directly on CX3CR1, but on EGFR [6,7]. CX3CR1 can activate the metalloproteinase which releases the EGFR activator. The activated EGFR directly stimulates the cancer cell to proliferate.

Another pathway activated by CX3CR1 is PI3K→ Akt/PKB, which activates NF-κB [117]. As a result, the expression of B-cell lymphoma 2 (Bcl-2) and B-cell lymphoma-extra large (Bcl-x_L_) increases, which causes apoptosis resistance in pancreatic cancer cells [12]. Moreover, the activation of NF-κB stimulates the proliferation of pancreatic cancer cells. The activation of this transcriptional factor may also cause an increase in CX3CL1 expression, which takes place in smooth muscle cells [49], where the expression of this chemokine acts in an autocrine manner.

## 6. The role of CX3CL1 in Apoptosis in a Tumor

Apoptosis is a consequence and at the same time a stage in the development of cancer. The high frequency of mutations and the intense proliferation of cancer cells lead to their genomic heterogeneity [118], which may result in their varied response to apoptotic conditions. In the early stages, the lack of development of blood vessels that would nourish the growing tumor results in hypoxia in the tumor, which triggers the apoptosis of those cancer cells most susceptible to adverse environmental conditions [92]. This is followed by an increase in the number of cancer cells with mutations that make them insensitive to apoptosis induction, e.g., with non-functional p53 [119,120].

The formation of apoptotic bodies intensifies tumor development [121,122] by inducing an anti-inflammatory effect and reducing the immune response [123]. Additionally, apoptotic bodies contain factors causing chemotaxis of phagocytic cells, mainly the chemotaxis of macrophages to the tumor niche. One example of such factors is sphingosine-1-phosphate (S1P) [124,125,126]. A second example is CX3CL1, which causes the chemotaxis of macrophages [127,128] and microglia [28,129]. CX3CL1 from apoptotic bodies can act on TAM and influence the distribution of these cells inside the tumor, which seems to be only local, although no thorough research has been conducted in this regard. In vivo studies show that TAM recruitment is mainly the responsibility of several representatives of the β-chemokine family: monocyte chemoattractant protein 1 (MCP-1)/CC motif chemokine ligand (CCL)2 [130,131,132,133], macrophage inflammatory protein 1β (MIP-1β)/CCL4 [130], regulated on activation, normally T cell expressed and secreted (RANTES)/CCL5 [134] and eotaxin-3/CCL26 [135].

CX3CL1 from apoptotic bodies also causes increased expression of milk fat globule epidermal growth factor VIII (MFG-E8) in macrophages [136]. This protein opsonizes apoptotic bodies, and thus, it participates in the removal of these particles by phagocytic cells, especially by macrophages [137]. Thanks to this, CX3CL1 increases the intensity of phagocytosis of apoptotic cells, and hence, increases the effect of these bodies on macrophages.

## 7. The Role of the CX3CL1-CX3CR1 Axis on Cancer Cell Migration and Metastasis

CX3CR1 is a chemokine receptor whose activation causes cell chemotaxis [45]. For this reason, an increase in CX3CR1 expression on cancer cells increases their migration [14,15,138]. The increase in CX3CR1 expression, a consequence of neoplastic processes, is further enhanced by hypoxia. The *CX3CR1* gene promoter has hypoxia response elements [139], which allows HIF-1 to bind to these sequences in hypoxia, thus increasing the expression of CX3CR1. This is particularly significant in ovarian cancer cells [140], pancreatic ductal adenocarcinoma cells [139] and prostate cancer cells [141]. Under hypoxic conditions CX3CR1 expression is also increased by NF-κB as confirmed by research on prostate cancer cells [141]. Hypoxia is also associated with an increase in the expression of ADAM10 [142] and TACE/ADAM17 [143,144,145], enzymes releasing sCX3CL1, which causes the activation of CX3CR1.

The *CX3CR1* gene promoter contains two Smad binding elements, thanks to which TGF-β increases the expression of CX3CR1 in microglia [146], although more detailed research on the regulation of CX3CR1 expression by TGF-β is required for different cancer cells. For example, one study showed that TGF-β does not change the CX3CR1 expression level in pancreatic ductal adenocarcinoma cells [20].

The increase in CX3CR1 expression increases sCX3CL1-induced neoplastic cell migration. sCX3CL1 may come from cancer cells [147] and also from fibroblasts, which secrete this chemokine from membrane microvesicles, as shown in prostate cancer models [148]. The activation of CX3CR1 activates multiple signaling pathways, in particular the Src → FAK pathway in breast cancer cells [14], lung cancer cells [16], pancreatic ductal adenocarcinoma cells [13] and in prostate cancer cells [15]. However, it seems that the Src → FAK pathway is only indirectly activated via CX3CR1. This receptor activates EGFR, which is directly responsible for the activation of the Src → FAK pathway and cancer cell migration, as shown on prostate cancer cells [15]. Apart from this pathway, CX3CR1-induced migration also crucially involves p38 MAPK, ERK MAPK and the PI3K → Akt/PKB axis [149].

Activation of CX3CR1 causes EMT in pancreatic ductal adenocarcinoma [48] and prostate cancer cells [52]. This effect depends on the activation of JAK2 → STAT3 in pancreatic ductal adenocarcinoma cells [48]. It has been shown that it is also dependent on the release of amphiregulins, HB-EGF and TGF-α, and the activations of EGFR/ErbB1 receptors on prostate cancer cells [52] and EGFR/ErbB1 and ErbB2 on breast cancer cells [7]. The EGFR family-dependent migration of cancer cells induced by CX3CR1 activation is probably a single signaling pathway, since the activation of STAT3 can take place via EGFR/ErbB1 [51]. However, further research on the exact mechanism and the role of EGFR family in EMT induced by CX3CR1 activation is required.

The activation of CX3CR1 also increases the expression of MMP-9 [15,111] and MMP-2 [111], metalloproteinases which participate in migration and angiogenesis.

The growth of the tumor is accompanied by the emergence of mechanisms that induce the migration of tumor cells. It is estimated that as many as 1 million cancer cells are released daily into the blood from just 1 g of cancer tissue [150]. In the bloodstream, the circulating cancer cells show low viability due to many factors, e.g., lack of attachment to the extracellular matrix, which causes anoikis [151]. However, circulating cancer cells are disseminated throughout the bloodstream, adhere to the blood vessels, and consequently, form a tumor in an organ other than that from which the cancer cells originate [152]. This process, known as metastasis, involves mCX3CL1 as one of the adhesion proteins. Due to the high expression of CX3CL1 in bones [14], lungs [29] and nervous tissues [25,26,27,28], circulating cancer cells with CX3CR1 expression cause metastasis in these organs and tissues (Figure 3). For this reason, breast cancer often causes metastases to spinal cancellous bone, bones [14,153,154] and brain [155], osteosarcoma to the lungs [117], while ovarian carcinoma is associated with peritoneal metastasis [6,21], pancreatic ductal adenocarcinoma causes perineural invasion [13,20,139,156] and prostate cancer causes bone metastasis [15,19,47]. Particularly in prostate cancer, high levels of dihydrotestosterone increase the cleavage of mCX3CL1 in bones [19], which causes a release of sCX3CL1 and the migration of prostate cancer cells to the bone marrow.

CX3CL1 can participate in so-called inflammatory oncotaxis, a process that involves the formation of metastasis at the site of inflammation caused by mechanical trauma [157,158]. Ischemia-reperfusion brain injury [159,160], skin wounds [161] and arterial injuries [162] induce greater production of CX3CL1. This process participates in the healing of damaged tissue. However, it can also take part in metastasis. The large majority of circulating cancer cells will not form a tumor because the subsequent stages of metastasis depend on many factors and are highly ineffective. Some of these factors are chemotactic agents secreted from wounds and adhesion proteins near damaged blood vessels, for example, mCX3CL1. Therefore, it can be assumed that this chemokine supports skin metastasis in wounds [163,164,165] or the migration of glioblastoma multiforme (GBM) cells to stroke areas [87,166]. At these sites, neoplastic cells find favorable conditions for further development and form a metastasis [167]. These are just general premises of the potential role of the CX3CL1-CX3CR1 axis in inflammatory oncotaxis, and need to be backed by experimental research.

The CX3CL1-CX3CR1 axis may also indirectly participate in organ-specific metastasis, where it may cause increased surface expression of CXC motif chemokine receptor 4 (CXCR4) on chronic lymphocytic leukemia cells [149]. This process is dependent on the activation of the PI3K → Akt/PKB pathway, but the exact mechanism has not yet been determined. In general, Akt/PKB activation increases CXCR4 expression at the transcription level [168,169], a process which may depend on Akt/PKB → NF-κB [170] or Akt/PKB → mTOR [171] depending on the experimental model. Therefore, it may be postulated that CX3CL1 causes an increase in CXCR4 expression through one of these pathways. The increased expression and activation of CXCR4 is necessary for bone marrow metastasis [172]—as the high expression of CXC motif chemokine ligand (CXCL)12 in the bone causes the homing of only those cells which have CXCR4 expression. Additionally, the activation of CX3CR1 on human myeloma RPMI-8226 cells increases their adhesion to fibronectin and vascular cell adhesion molecule-1 (VCAM-1) [173], which is associated with the migration of these cells to bone marrow.

CX3CL1-CX3CR1 axis stimulates cancer cell migration, although it can also inhibit migration. As mCX3CL1 is an adhesion protein for cells with CX3CR1 expression, cancer with a simultaneous expression of both these proteins may aggregate, and thus, the migration of these cells is prevented [79]. This process is significant in a GBM tumor [87], characterized by high expressions of CX3CR1 [174,175,176] and CX3CL1 [147]. CX3CL1 expression is inversely controlled by TGF-β [87], with a higher concentration of TGF-β reducing the expression of CX3CL1 and increasing the migration of GBM cells.

## 8. The Role of CX3CL1 in Angiogenesis: Influence on Endothelial Cells

Blood vessels do not grow commensurately with the tumor, which inevitably results in hypoxia in the tumor, triggering reactions that stimulate angiogenesis [177,178,179], a complex process in the tumor microenvironment involving many signal particles and cell types [180]. Two of the most significant are vascular endothelial growth factor (VEGF) [181] and TAM [182]. There are also many smaller signal molecules that participate in angiogenesis in the tumor. One such molecule is CX3CL1. Studies show that the CX3CL1-CX3CR1 axis is important in the angiogenesis of tumors such as breast cancer [183], hepatocellular carcinoma [184], lung cancer [185], malignant melanoma [186] and multiple myeloma [187].

CX3CL1-CX3CR1 influences angiogenesis in two ways. CX3CL1 participates in the recruitment of pro-angiogenic TAM [183,188]. CX3CL1 may also directly affect endothelial cells, causing their proliferation, migration and tube formation [9,10,11] (Figure 4). The latter process may intensify in hypoxia, a state shown to increase the expression of CX3CL1 in human umbilical vein endothelial cells (HUVEC) [189] and prostate cancer cells [8]. The exact mechanisms of the effect of CX3CL1 on endothelial cells vary. In human aortic endothelial cells, activation of CX3CR1 causes the activation of ERK MAPK, followed by an increase in HIF-1α protein expression and the production of VEGF-A_165_ but not VEGF-D [10]. Then, VEGF-A_165_ causes angiogenesis. In contrast, a study on HUVEC showed that CX3CL1 does not cause VEGF-dependent angiogenesis. CX3CL1 at 10 ng/mL (0.33 nM) activated two pathways in those cells: ERK MAPK and PI3K → Akt/PKB → endothelial nitric oxide synthase (eNOS) → nitric oxide (NO) [9]. This resulted in the stimulation of proliferation, migration and tube formation of HUVEC. However, at a concentration of 300 ng/mL (10 nM), CX3CL1 in HUVEC and human microvascular endothelial cells did not activate Akt/PKB but only c-Jun N-terminal kinase (JNK) and ERK MAPK, MAPK kinases responsible for the migration of endothelial cells [11]. Significantly, the concentration of 300 ng/mL (10 nM) is much higher than the physiological level [190], and thus, this model may not show the real mechanisms in which CX3CL1 participates. Another problem is the participation of EGFR/ErbB1 in signal transduction from CX3CR1. In a research model of primary human coronary artery smooth muscle cells, CX3CR1 directly activated ERK MAPK via G_i_. However, the activation of the PI3K→ Akt/PKB pathway was dependent on EGFR/ErbB1 [50] and the shedding of epiregulin. Probably, in endothelial cells, CX3CL1 causes the activation of EGFR/ErbB1; this receptor directly participates in angiogenesis.

## 9. Recruitment of Macrophages by the CX3CL1-CX3CR1 Axis

Tumors also contain macrophages, which are recruited blood monocytes [182]. During the recruitment process, monocytes are differentiated into macrophages with specific characteristics. Intensive inflammatory reactions result in the differentiation of monocytes into M1 phenotype macrophages [191], pro-inflammatory macrophages with low CX3CR1 expression. Accumulation of these cells in the tumor improves the prognosis of patients with glioma [192]. However, in the tumor there are also factors inhibiting immune reactions, resulting in the differentiation of monocytes into M2 phenotype macrophages and TAM [191]. Macrophages with these two phenotypes have a high expression of CX3CR1 [191], and a high percentage of these cells is associated with a poor prognosis [192]. However, it seems that the polarization of macrophages is flexible [193] and largely depends on factors affecting the macrophages at a given moment.

Monocytes are recruited to the tumor niche by chemokines. Most often, these cells are recruited by MCP-1/CCL2 [130,131,132,133,194], MIP-1β/CCL4 [130] and RANTES/CCL5 [134,194]. Additionally, the recruitment of monocytes through CX3CL1 occurs in in vivo models of such cancers as human breast cancer line HS578T [183], human colorectal adenocarcinoma line DLD1 [18] and human colorectal carcinoma line HCT116 [18]. It seems that this CX3CL1-dependent recruitment occurs only in some cancers. For example, a tumor produces a variety of chemokines that induce the recruitment of monocytes [176]. CX3CL1 is only one of them and often does not play a major role in the process.

Recruitment of monocytes to a tumor involving the CX3CL1-CX3CR1 axis is performed under strictly defined conditions (Figure 5). Three populations of monocytes in blood are distinguished: classical monocytes (CD14^++^CD16^-^), intermediate monocytes (CD14^++^CD16^+^) and non-classical monocytes (CD14^+^CD16^++^) [195,196]. Classical monocytes have a low expression of CX3CR1, but high expressions of CC motif chemokine receptor (CCR)2 and CCR5. In turn, non-classical monocytes have high CX3CR1 expression and low CCR2 and CCR5 expression. Therefore, these two monocyte populations are recruited by different chemokines [197]. Compared to classical monocytes, non-classical monocytes under the influence of LPS produce more pro-inflammatory cytokines but lower levels of interleukins 10 (IL-10) and chemokines MCP-1/CCL2 and RANTES/CCL5 [195]. Additionally, after differentiation to monocyte-derived dendritic cells, non-classical monocytes activate CD4^+^ T cells more strongly (i.e., than those originating from classical monocytes) [198]. However, non-classical monocytes can be recruited to a tumor by CX3CL1, where they perform proangiogenic roles [18]. Although sCX3CL1 acts chemotactically on non-classical monocytes, CX3CR1 binds with both forms of CX3CL1, and so at a high concentration of sCX3CL1, adhesion of non-classical monocytes to mCX3CL1 is impaired [199]. However, sCX3CL1 increases adhesion to fibronectin and intracellular adhesion molecule-1 (ICAM-1) [200].

There is also a second mechanism preventing the recruitment of pro-angiogenic non-classical monocytes to zones with intensive inflammatory reactions. During these reactions, TNF-α is released by the non-classical monocytes, but this requires a proteolytic cleavage by TACE/ADAM17 [201], an enzyme that also mediates the shedding of mCX3CL1, which prevents monocyte adhesion to this protein [202].

Another mechanism of the influence of sCX3CL1 on monocyte migration is by interference with the activity of other chemokines. Without the presence of sCX3CL1, MCP-1/CCL2 causes an increase in monocyte adhesion, during which an increase in CX3CR1 surface expression may occur, resulting in improved adhesion to mCX3CL1 [203]. However, sCX3CL1 antagonizes MCP-1/CCL2-dependent trans-endothelial migration and chemotaxis of classical monocytes [204]. It seems that such an interaction between these chemokines occurs only in monocytes. CX3CL1-dependent recruitment of NK cells is not disturbed by MCP-1/CCL2 [204].

Inflammatory reactions increase mCX3CL1 levels on endothelial cells [18,30,31,32,33]. This leads to the adhesion of non-classical monocytes to vessel walls near the tumor [18], followed by the production of IL-6, MCP-1/CCL2 and MMP-9 by monocytes [205]. However, the high levels of mCX3CL1 do not allow trans-endothelial migration of these monocytes, and thus, their retention occurs on the endothelium [197]. For trans-endothelial migration of the monocytes to occur, it is necessary to reduce the level of mCX3CL1 on the endothelial cells, which takes place in hypoxia [31] and under the influence of TGF-β [54]. It seems that this process in the tumor involves VEGF-A [18], a growth factor causing GATA3 to attach to the *CX3CL1* gene, and thus, reducing its expression. In this way, it enables the trans-endothelial migration of non-classical monocytes.

Monocytes in the tumor niche differentiate into TAM, which shows CX3CR1 expression. The activation of this receptor inhibits the apoptosis of TAM cells [188], which results in an increase in their number in the tumor niche where they act pro-angiogenically [183]. Among other things, these TAM secrete MMP-9 which supports angiogenesis [18]. Additionally, in the tumor niche it is possible that CX3CL1 increases the production of platelet-derived factor 4 (PF-4)/CXCL4 in macrophages [206]. PF-4/CXCL4 amplifies VEGF-mediated angiogenesis. However, the direct influence of CX3CL1 on the pro-angiogenic properties of macrophages still needs to be thoroughly investigated, because CX3CL1 has been shown to reduce the production of VEGF-A by the granulocyte-macrophage colony-stimulating factor (GM-CSF)-stimulated macrophages, but at the same time increase expression of anti-angiogenic thrombospondin 1 (THBS-1) and disintegrin, and metalloproteinase with a thrombospondin motif (ADAMTS-1) [207], which inhibits angiogenesis.

The CX3CL1-CX3CR1 axis also participates in cellular interactions in the tumor niche. In a tumor, TAM shows a high expression of CX3CR1 [191,208]. The production of CX3CL1 [147] by cancer cells enables TAM migration to the vicinity of these cells [209], followed by the cross-talk of TAM and cancer cells through intercellular signaling. IL-10 produced by macrophages induces CX3CR1 expression on lung cancer cells [185]. Additionally, I-309/CCL1, macrophage inflammatory protein 1α (MIP-1α)/CCL3 and granulocyte colony-stimulating factor (G-CSF) from TAM increase CX3CL1 expression on lung cancer cells [185]. The increase in CX3CL1-CX3CR1 axis expression participates in migration, invasion, metastasis and angiogenesis of lung cancer [185].

The CX3CL1-CX3CR1 axis has important functions in the brain. CX3CR1 expression occurs in microglia [25,26,27], while nerve cells have CX3CL1 expression, which enables microglia chemotaxis to the nerve cells. A similar mechanism takes place between microglia and NF1-optic pathway gliomas [210]. In the GBM tumor, microglia group around the tumor mass and then infiltrate the tumor [211]. However, the recruitment of microglia to a GBM tumor does not depend on the CX3CL1-CX3CR1 axis [212,213], although one study on the polymorphism of the *CX3CR1* gene showed that the CX3CL1-CX3CR1 axis is important for the recruitment of tumor tissue by microglia [214]. Significantly, CX3CL1 is not the only chemokine involved in this process, e.g., also important in the recruitment of microglia to GBM tumor are MCP-1/CCL2 [132] and RANTES/CCL5 [215].

The CX3CL1-CX3CR1 axis seems to work only locally. TAM and microglia show high expression of CX3CR1 [132,208,209], whereas high CX3CL1 expression occurs in other cells [209], e.g., in GBM cells [147], especially in cells with IDH1-R132H mutation of the gene encoding isocitrate dehydrogenase 1 (*IDH1*) [78]. This results in the phosphorylation of NF-κB and an increase in CX3CL1 expression. CX3CL1 expression enables the migration and adhesion of TAM and microglia to GBM cells [209], which increases the intercellular signaling between these cells, although at the same time it also causes the migration and accumulation of NK cells [78].

CX3CL1 reduces inflammatory reactions in macrophages [133,190] and microglia [213,216,217], although this effect may depend on the microenvironment, because in hypoxia, CX3CR1 is important for the activation of microglia [218]. The anti-inflammatory effect of CX3CL1 also depends on its concentration and type of cells. In macrophages, CX3CL1 at 0.9 ng/mL (0.03 nM) reduced inflammatory reactions [190] via a mechanism dependent on the increased level of anti-inflammatory prostaglandin 15-deoxy-Δ^12,14^-prostaglandin J_2_ (15d-PGJ_2_) [219]. However, at 90 ng/mL (3 nM) in macrophages, CX3CL1 increased the level of interleukins 23 (IL-23), which offsets its anti-inflammatory effect [190]. The concentration of 90 ng/mL (3 nM) occurs in inflammatory reactions, and thus, the nature of the action of CX3CL1 on inflammatory reactions in macrophages in a tumor depends on the intensity of these reactions. In contrast, in microglia, CX3CL1 at 200 ng/mL (6 nM) had an anti-inflammatory effect [217].

The suppression of inflammatory reactions by CX3CL1 results in reduced production of IL-1β [213,216,217] and TNF-α [190,216], and increased expression of heme oxygenase 1 (HO-1) [133]. It has been reported that an increase in HO-1 expression in gut macrophages protects against colorectal carcinogenesis [133]. Additionally, in GBM, a reduction in IL-1β expression by CX3CL1 has an anticancer effect [213] because IL-1β is important in GBM cell proliferation and migration [220].

## 10. Impact on the Recruitment of Myeloid-Derived Suppressor Cells

MDSC are responsible for tumor immune evasion in a tumor niche [221]. This heterogeneous group of cells includes monocytic myeloid-derived suppressor cells (Mo-MDSC) with high expression of CX3CR1 [222], which allows them to accumulate in a tumor with a high expression of sCX3CL1 [17]. Due to the expression of sCX3CL1 by a tumor, Mo-MDSC can migrate close to these cells and offset the effects of anticancer lymphocytes [147]. However, eotaxin-3/CCL26 is a chemokine that, in addition to activating its CCR3 receptor [135,223], can also activate CX3CR1 [224]. For this reason, Mo-MDSC may also be recruited by the eotaxin-3/CCL26 → CX3CR1 axis [225].

Mo-MDSC recruitment by the two chemokines takes place mainly during hypoxia, when the expression of eotaxin-3/CCL26 [225] and CX3CL1 [8,52] increases in cancer cells. In the tumor, Mo-MDSC participates in tumor immune evasion, causing T cell anergy by producing NO [226]. Mo-MDSC also produces chemokines that cause the recruitment of T_reg_ [227], cells that also participate in tumor immune evasion [228]. Mo-MDSC also has a high expression of MMP-9, an enzyme that supports migration, invasion and angiogenesis in the tumor [225].

## 11. The Importance of CX3CL1 in Cancer Processes Involving HCMV

In many types of tumors, increased expression of CX3CL1 causes an increased immune response against these tumors, and thus, a better prognosis for the patient [73,79,81]. However, some cancers show the opposite correlation. For example, increased expression of CX3CL1 gives a poorer prognosis in grade III-IV brain tumors [110]. One of the possible causes is the effect of CX3CL1 on the human cytomegalovirus (HCMV), because this chemokine plays an important role in the development cycle of this virus, which in turn may have an important function in GBM. However, the connection between HCMV and GBM is a highly contentious issue. Some researchers have shown that the genetic material of the virus and its antigens are present in the GBM tumor [229,230,231]. It has also been confirmed that HCMV theoretically intensifies cancer processes [232,233]. For this reason, immunotherapy against HCMV antigens [234] or antiviral drugs [235] give good results in GBM treatment. On the other hand, some researchers have demonstrated no active HCMV infection in the GBM tumor [236,237].

HCMV occurs in about 60% of the population [238,239]. In people with a normally functioning immune system, HCMV infection is almost always asymptomatic and leads to a latent infection of some cells. HCMV causes latent infection of CD34^+^ hematopoietic progenitor cells [240,241], which become the reservoir of this virus, passing it to daughter cells, including monocytes. However, HCMV is unable to replicate itself in these cells [242], and thus, the monocyte infection is latent. In these cells, the virus is spread throughout the body [243]. An important element of the latent infection is the expression of US28 [240,244,245], a protein also important for the migration of infected monocytes. US28 is a non-specific chemokine receptor [246] associated with CX3CL1 [247]. The activation of US28 by CX3CL1 increases the migration of infected cells [246] and this effect is greater than the activation of CX3CR1 by CX3CL1 [248]. Due to this, US28 participates in the adhesion of latent infected monocytes, transport of these cells through the vessel walls, and their recruitment to sites with a high expression of mCX3CL1 [249,250]. One such site is a tumor, a place of intensive immunological reactions and high expression of CX3CL1. This environment with immunological defects is convenient for the active replication of the virus. US28 may also be involved in cell infection [251], e.g., by chemotaxis of infected cells to cells showing sCX3CL1 expression. However, these processes are very difficult to study because US28 is only found in the human strains of cytomegalovirus [249].

## 12. Conclusions: The Application of CX3CL1 in Cancer Therapy

An increased expression of CX3CL1 in a cancer cell leads to the infiltration of NK cells, dendritic cells, CD4^+^ and CD8^+^ T cells into the tumor [61,63,69,70,71,72,73,74,75,76,77,78]. This, in turn, leads to an increase in the anti-tumor immune response, which reduces the rate of tumor growth and increases the survival of experimental animals and cancer patients. Therefore, gene therapy with the use of the adenoviral vector [61], plasmid vector or DNA vaccine [72,252,253] with the CX3CL1 gene that increases the expression of CX3CL1 in the tumor has a therapeutic effect. In addition, it is postulated to simultaneously increase the expression of CX3CL1 and other cytokines that enhance the immune response, e.g., IL-2, which significantly increases the effect of CX3CL1 in a tumor [252]. Additionally, increased expression of CX3CR1 ex vivo in T cells achieved by the use of the retrovirus vector, and introducing these cells into the body of mice has led to an increase in the homing of these cells to a colorectal tumor [254], which was associated with a high CX3CL1 expression in the tumor and had an anti-tumor effect. Another study showed that the use of ex vivo adenovirus vector in increasing CX3CL1 expression in dendritic cells and introducing these cells into a tumor increased the anti-tumor response [255], in particular the homing of CD8^+^ T cells and CD4^+^ T cells to the tumor.

However, therapy with this chemokine alone will be ineffective due to the numerous processes in a tumor, which results in cancer immune evasion [256]. In addition, some cancers enhance the pro-cancer properties of CX3CL1. Currently, it is postulated that immunotherapy should be combined with drugs that suppress the immunosuppressive mechanisms in a tumor [257]. As such, CX3CL1 may be used as one of the elements of immunotherapy, an adjuvant increasing the effectiveness of anticancer therapy.

## Figures and Tables

**Figure 1 ijms-21-03723-f001:**
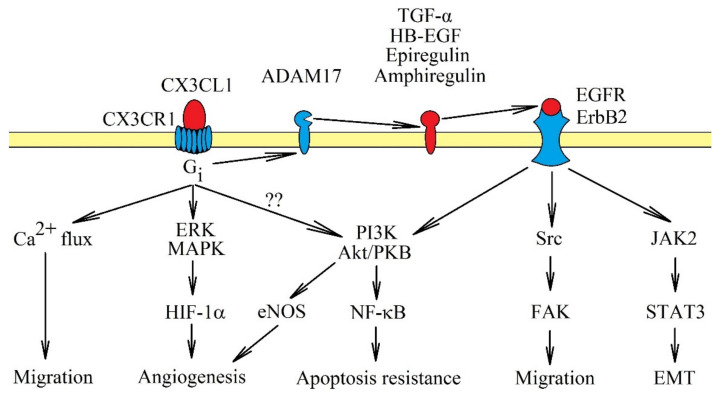
Signal transduction from CX3C chemokine receptor 1 (CX3CR1). The CX3CR1 receptor is a seven-transmembrane domain G protein-coupled receptor. Activation of this receptor causes signal transmission to extracellular signal-regulated kinase (ERK) mitogen-activated protein kinase (MAPK) and phosphatidylinositol-4,5-bisphosphate 3-kinase (PI3K) → Akt/protein kinase B (PKB) axis, accompanied by Ca^2+^ mobilization. These pathways cause cancer cell migration and apoptosis resistance. CX3CR1 activation also activates Src → focal adhesion kinase (FAK) and janus tyrosine kinase 2 (JAK2) → signal transducer and activator of transcription 3 (STAT3), although it is likely that they are only activated indirectly. First, CX3CR1 activates tumor necrosis factor-α converting enzyme/a disintegrin and metalloproteinase 17 (TACE/ADAM17), which releases epidermal growth factor receptor (EGFR)/ErbB1 and ErbB2 receptor activators. Only after the activation of EGFR/ErbB1 and ErbB2 receptors, can Src → FAK and JAK2 → STAT3 pathways be activated as well. CX3CL1: CX3C chemokine ligand 1.

**Figure 2 ijms-21-03723-f002:**
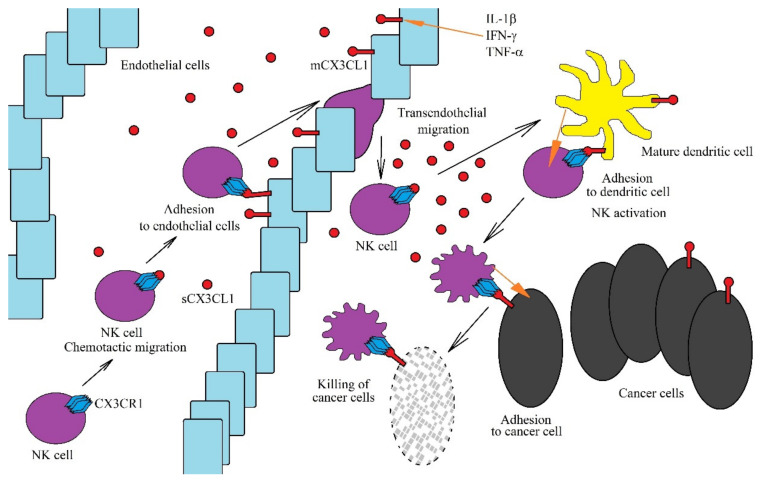
Role of the CX3CL1-CX3CR1 axis in anti-cancer NK cells functions. The CX3CL1-CX3CR1 axis plays an important role in the anti-tumor functions of NK cells at many levels. CX3CL1 expression is increased by the proinflammatory cytokines IL-1β, IFN-γ and TNF-α. A sCX3CL1 gradient is formed, which causes chemotactic migration of NK cells. On endothelial cells near the tumor, mCX3CL1 acts as an adhesive protein for NK cells. This enables the transendothelial migration of NK cells to sites of intensive immune response, where they are activated by mature dendritic cells. However, for NK cells to be activated effectively, chemotaxis and adhesion of these cells to mature dendritic cells is required, which is dependent on the CX3CL1-CX3CR1 axis. Activated NK cells destroy cancer cells. However, in order to do this more effectively, they must adhere to cancer cells via the CX3CL1-CX3CR1 axis.

**Figure 3 ijms-21-03723-f003:**
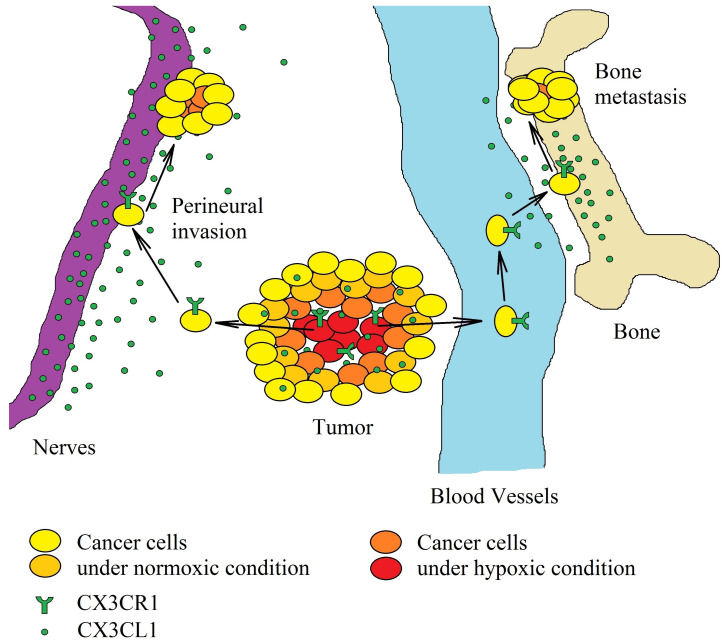
Significance of the CX3CL1-CX3CR1 axis in hypoxia-induced metastasis. Hypoxia induces an increase in CX3CR1 expression on cancer cells. The increased concentration of CX3CL1 in the tumor causes the migration of tumor cells. A neoplastic cell with CX3CR1 expression migrates to areas with a high CX3CL1 concentration within the same organ. Because neurons are one of the sources of this chemokine, perineural invasion may occur, often in pancreatic ductal adenocarcinoma. In the bloodstream, on the other hand, cancer cells with CX3CR1 expression migrate to the brain and bones—organs with a high expression of CX3CL1.

**Figure 4 ijms-21-03723-f004:**
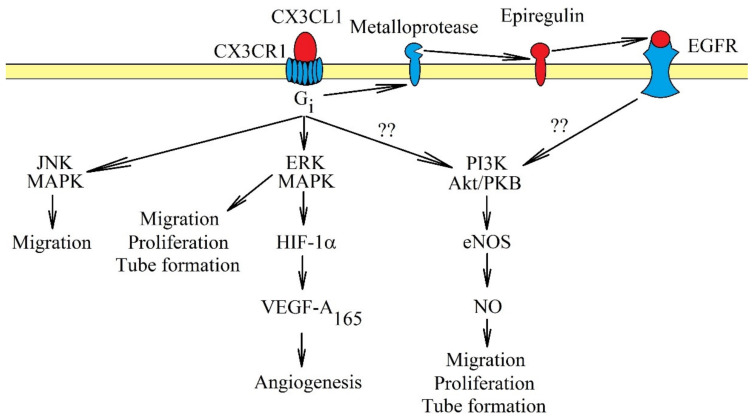
Intracellular signaling in endothelial cells in the induction of angiogenesis by CX3CL1. Activation of CX3CR1 causes signal transduction in endothelial cells, which causes angiogenesis. The most important activated pathways in this process are ERK MAPK and PI3K → Akt/PKB. Activation of ERK MAPK causes migration, proliferation and tube formation of endothelial cells. This MAPK cascade may also participate indirectly in angiogenesis and increase the expression of HIF-1α, responsible for increasing VEGF-A_165_ production. This growth factor participates directly in angiogenesis. Another pathway activated by CX3CR1 is PI3K → Akt/PKB, which in turn increases the activation of eNOS and NO production. It is likely that it is activated directly by EGFR, and CX3CR1 plays only an indirect role. Another important pathway in angiogenesis caused by CX3CR1 activation is JNK MAPK. Activation of this cascade causes migration of endothelial cells.

**Figure 5 ijms-21-03723-f005:**
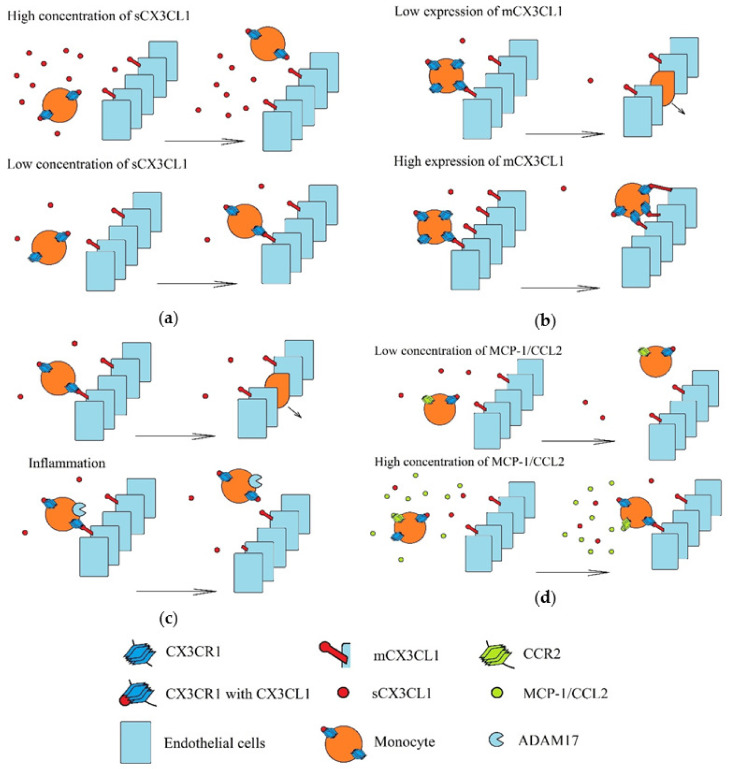
Recruitment of monocytes in various environments. Monocytes are recruited via mCX3CL1-dependent adhesion to vascular walls and transendothelial migration. These processes depend on: (**a**) the concentration of sCX3CL1. A high concentration of this chemokine activates all CX3CR1 and it can no longer bind more CX3CL1, including mCX3CL1. (**b**) Concentrations of mCX3CL1 on the vascular walls. A high concentration of mCX3CL1 causes very strong monocyte adhesion, which causes retention of these cells on the vascular walls and inhibit transendothelial migration of monocytes. (**c**) Inflammatory processes induced in recruited monocytes, in particular the level of TACE/ADAM17 activity. Inflammatory reactions increase the production of TNF-α and TACE/ADAM17 (an enzyme that releases this pro-inflammatory cytokine). However, this metalloproteinase also cleaves mCX3CL1. This results in the releases of sCX3CL1, and hence, reduces the amount of mCX3CL1 on the walls of the blood vessels. (**d**) Concentrations of other chemokines, in particular MCP-1/CCL2. Activation of MCP-1/CCL2 receptor (CCR2) causes increase CX3CR1-dependent monocyte adhesion.

**Table 1 ijms-21-03723-t001:** Prognosis for various types of cancer with increased expression of CX3CL1 or CX3CR1 according to reports available in PubMed.

Type of Cancer	Prognosis at Increased Expression of a Given Protein in Tumor	Number of Patients in the Study	Comments	Source
**CX3CL1**				
Colorectal cancer	↓	174	Plasma samples	[113]
Colorectal Cancer	↑	100	Co-expression of CX3CL1 and CX3CR1	[79]
Colorectal Cancer	↑	50		[71]
Breast cancer	↓	753		[76]
Breast carcinoma	↑	204		[74]
Gastric adenocarcinoma	↑	158		[73]
Glioma	↑	61		[78]
Grades III–IV brain tumours	↓	36		[110]
Hepatocellular carcinoma	↑	56	Co-expression of CX3CL1 and CX3CR1	[109]
Lung adenocarcinoma	↑	--	From The Cancer Genome Atlas	[80]
Lung adenocarcinoma	↓	41	Patients with smoking history	[111]
Pancreatic ductal adenocarcinoma	↓	105		[112]
Soft tissue sarcomas	↑	69	Female	[81]
**CX3CR1**				
Clear cell renal cell carcinoma	↓	78		[46]
Colorectal Cancer	↑	100	Co-expression of CX3CL1 and CX3CR1	[79]
Hepatocellular carcinoma	↑	56	Co-expression of CX3CL1 and CX3CR1	[109]
Epithelial ovarian carcinoma	↓	557		[21]
Pancreatic ductal adenocarcinoma	↓ *p* = 0.059	105		[112]
Pancreatic ductal adenocarcinoma	↑	104		[20]

↑—better prognosis; ↓—worse prognosis; --—no correlation

**Table 2 ijms-21-03723-t002:** Prognosis for various types of cancer with increased expression of CX3CL1 or CX3CR1 according to The Human Protein Atlas [114].

Type of Cancer	Prognosis with Increased Expression of CX3CL1 in the Tumor	Prognosis with Increased Expression of CX3CR1 in the Tumor
Glioma	--	↑ *p* = 0.060
Thyroid cancer	↑	↑
Lung cancer	↑	↑
Colorectal cancer	↓	--
Head and neck cancer	--	↑
Stomach cancer	↓ *p* = 0.052	↓
Liver cancer	↓ *p* = 0.071	--
Pancreatic cancer	↓	↑
Renal cancer	↑	↑
Urothelial cancer	↓ *p* = 0.093	--
Prostate cancer	↑	--
Testis cancer	--	↓ *p* = 0.086
Breast cancer	↑	--
Cervical cancer	↑	↑ *p* = 0.051
Endometrial cancer	↑	↑
Ovarian cancer	--	↓
Melanoma	↑	↑

↑—better prognosis; ↓—worse prognosis; --—no correlation

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
