# Peer review of "Fractalkine/CX3CL1 in Neoplastic Processes"

_ijms, 2020, doi:10.3390/ijms21103723_

Round 1

Reviewer 1 Report

This review broadly summarized the current knowledge about CX3CL1/CX3CR1 axis on neoplasm and cancer metastasis. It will be useful for gaining insight into tumor microenviroment and metastasis.

  1. Regarding the prognosis for human patients, the expression level of CX3CL1 or its receptors alone gives the opposite consequences in different cancers. It will be interesting to talk about the association of co-expression level of CX3CL1 and CX3CLR1 with prognosis since the simultaneous expression of both these proteins on cancer cells may aggregate and prevent migration.
  2. CX3CL1 is highly enriched in the brain and brain metastasis is involved in many types of cancers. It will be nice to add more information about CX3CL1/CX3CR1 axis and brain metastasis other than GBM.
  3. It is well documented that CX3CL1/CX3CR1 axis participates in the regulation of immune cells within tumors including DCs, NKs and macrophages. Are there any studies related to the correlation of CX3CL1/CX3CR1 axis with the regulation of T cell activity within tumor?
  4. In the line 323, please provide more details about how CX3CL1/CX3CR1 axis regulates CXCR4 expression.

Author Response

This review broadly summarized the current knowledge about CX3CL1/CX3CR1 axis on neoplasm and cancer metastasis. It will be useful for gaining insight into tumor microenviroment and metastasis.

  1. Regarding the prognosis for human patients, the expression level of CX3CL1 or its receptors alone gives the opposite consequences in different cancers. It will be interesting to talk about the association of co-expression level of CX3CL1 and CX3CLR1 with prognosis since the simultaneous expression of both these proteins on cancer cells may aggregate and prevent migration.

The association of CX3CL1 expression with CX3CR1 is a more complicated issue because the expression of CX3CR1 may not be on cancer cells but on cells of the immune system. Then it indicates an active immune response. Another problem is the occurrence of CX3CL1 in two forms that, in the context of increasing CX3CR1 expression on a tumor cell, will theoretically have the opposite effect. Therefore, we have written an additional paragraph about the association of CX3CR1 expression with CX3CL1 expression in the tumor and the effect on patient prognosis.

  1. CX3CL1 is highly enriched in the brain and brain metastasis is involved in many types of cancers. It will be nice to add more information about CX3CL1/CX3CR1 axis and brain metastasis other than GBM.

We have added more information about CX3CL1/CX3CR1 axis and brain metastasis other than GBM. We also show that various organs, including nerve tissue, have high CX3CL1 expression which causes site-specific metastasis.

  1. It is well documented that CX3CL1/CX3CR1 axis participates in the regulation of immune cells within tumors including DCs, NKs and macrophages. Are there any studies related to the correlation of CX3CL1/CX3CR1 axis with the regulation of T cell activity within tumor?

The paragraph on the effect on CX3CL1 on the cells of the immune system. We have added information on T cells with a particular emphasis on CD8 T cells.

  1. In the line 323, please provide more details about how CX3CL1/CX3CR1 axis regulates CXCR4 expression.

The paragraph on the CX3CL1 effect on CXCR4 expression has been extended, even though the phonomenon itself is not well researched. In many studies it was shown that other factors also increase the mRNA expression of CXCR4 via Akt/PKB. However, it is not known what occurs after that Akt/PKB. It may be postulated that NF-kB or mTOR. However, it is not known what is next in the Act / GDP. It can be postulated that NF-kB or mTOR.

Reviewer 2 Report

A comprehensive review of CX3CL1 and its role in cancer. Very well written and would be of high interest to the field. The final section on immunotherapy could be expanded a bit to include how the CX3CL1 would be delivered? oncolytic virus, direct injection etc. 

Author Response

A comprehensive review of CX3CL1 and its role in cancer. Very well written and would be of high interest to the field. The final section on immunotherapy could be expanded a bit to include how the CX3CL1 would be delivered? oncolytic virus, direct injection etc.

In line with the Reviewer's suggestions, the passage on the application of CX3CL1 has been extended.